# IoT Water Quality Monitoring and Control System in Moving Bed Biofilm Reactor to Reduce Total Ammonia Nitrogen

**DOI:** 10.3390/s24020494

**Published:** 2024-01-12

**Authors:** Putu A. Suriasni, Ferry Faizal, Wawan Hermawan, Ujang Subhan, Camellia Panatarani, I Made Joni

**Affiliations:** 1Department of Physics, Faculty of Mathematics and Natural Science, Padjadjaran University, Jalan Raya Bandung-Sumedang KM 21, Sumedang 45363, West Java, Indonesia; putu21005@mail.unpad.ac.id (P.A.S.); ferry.faizal@unpad.ac.id (F.F.); c.panatarani@phys.unpad.ac.id (C.P.); 2Functional Nano Powder University Center of Excellence (FiNder U-CoE), Padjadjaran University, Jalan Raya Bandung-Sumedang KM 21, Sumedang 45363, West Java, Indonesia; wawan.hermawan@unpad.ac.id (W.H.); ujang.subhan@unpad.ac.id (U.S.); 3Department of Biology, Faculty of Mathematics and Natural Science, Padjadjaran University, Jalan Raya Bandung-Sumedang KM 21, Sumedang 45363, West Java, Indonesia; 4Department of Fisheries, Faculty of Fisheries and Marine Science, Padjadjaran University, Jalan Raya Bandung-Sumedang KM 21, Jatinangor, Sumedang 45363, West Java, Indonesia

**Keywords:** RAS, biofilter, MBBR, TAN, bubble diffuser, Internet of Things (IoT)

## Abstract

Traditional aquaculture systems appear challenged by the high levels of total ammoniacal nitrogen (TAN) produced, which can harm aquatic life. As demand for global fish production continues to increase, farmers should adopt recirculating aquaculture systems (RAS) equipped with biofilters to improve the water quality of the culture. The biofilter plays a crucial role in ammonia removal. Therefore, a biofilter such as a moving bed biofilm reactor (MBBR) biofilter is usually used in the RAS to reduce ammonia. However, the disadvantage of biofilter operation is that it requires an automatic system with a water quality monitoring and control system to ensure optimal performance. Therefore, this study focuses on developing an Internet of Things (IoT) system to monitor and control water quality to achieve optimal biofilm performance in laboratory-scale MBBR. From 35 days into the experiment, water quality was maintained by an aerator’s on/off control to provide oxygen levels suitable for the aquatic environment while monitoring the pH, temperature, and total dissolved solids (TDS). When the amount of dissolved oxygen (DO) in the MBBR was optimal, the highest TAN removal efficiency was 50%, with the biofilm thickness reaching 119.88 μm. The forthcoming applications of the IoT water quality monitoring and control system in MBBR enable farmers to set up a system in RAS that can perform real-time measurements, alerts, and adjustments of critical water quality parameters such as TAN levels.

## 1. Introduction

Global consumption of aquatic food has increased in recent years. The consumption of aquatic food grew from an average of 9.9 kg/capita in 1960 to 20.5 kg in 2019 and kept growing despite the decline to 20.2 kg/capita in 2020. The Blue Food Assessment (BFA), projects that the consumption of aquatic food will increase by 15 percent to supply, on average, 21.4 kg per capita in 2030 [1]. However, the increased aquatic food consumption does not align with the fisheries’ resources. The decline is due to overfishing, poor management, and biologically unsustainable stock. One solution to maintaining aquaculture stocks’ sustainability is using a recirculating aquaculture system (RAS). The RAS is a closed system that combines treatments and reuses the water, with less than 10% of the total water volume replaced daily [2]. One factor that affects water quality and guarantees the aquaculture species in RAS is the minimum amount of total ammonia nitrogen (TAN). TAN is produced from fish respiration and the decomposition of organic matter [3]. So, a biofilter is needed to ensure a minimum amount of TAN through the biological process. In this process, ammonium is oxidized to nitrite and then nitrate with the help of nitrifying bacteria [4]. The nitrification in biofilters is made possible by the autotrophic bacteria ammonia-oxidizing bacteria (AOB) and nitrite-oxidizing bacteria (NOB) [5].

A moving bed biofilm reactor (MBBR) is a biofilter where nitrifying bacteria are attached to the surface of the suspended carrier media and then form biofilm [6]. The nitrifying activity in MBBR is influenced by the availability of oxygen concentration in the water [7]. The oxygen diffusion from water to biofilm is the factor that affects the process in MBBR and consequently determines biofilm composition [8]. A bubble diffuser is a recent technology to provide a sufficient amount of oxygen. The diffuser maintains the dissolved oxygen (DO) to satisfy the oxygen demand in the system. Other water quality parameters affecting nitrifying bacteria in MBBR are pH, temperature, and total dissolved solids (TDS). Thus, it is crucial to maintain a safe range of water quality parameters to achieve an efficient nitrification process of the MBBR. Consequently, a real-time measurement device is required to track the water quality of the MBBR.

The Internet of Things (IoT) has recently gained much interest in real-time data measurement for cost-effective aquaculture management. IoT refers to smart devices or sensors that are uniquely addressable based on their communication protocols, are adaptable, and are autonomous with inherent security [9]. IoT water quality monitoring uses intelligent sensors and wireless communication to measure and analyze various water quality parameters, such as temperature, pH, turbidity, dissolved oxygen, etc. IoT water quality monitoring has many benefits, such as real-time data collection, remote access, automation, cost-effectiveness, and scalability [10,11,12]. However, IoT water quality monitoring also faces many challenges and limitations, such as data security, communication coverage, energy consumption, sensor calibration, data quality, and interoperability [13,14]. Moreover, IoT water quality monitoring applications may have different requirements and constraints, such as water source, environment, types of sensors, purpose, and stakeholders [15,16]. Therefore, there is no one-size-fits-all solution for IoT water quality monitoring, and different methods, techniques, models, systems, and applications need to be compared and contrasted based on their strengths and weaknesses, as well as their suitability and feasibility for different scenarios.

The IoT’s architecture comprises hardware consisting of sensor nodes and middleware consisting of data storage and a presentation layer. The presentation layer should give efficient visualization and be compatible with various application platforms [17]. A study was conducted to monitor the water quality of freshwater aquaculture by integrating temperature, pH, DO, and EC sensors. The sensors are connected to the ESP 32 Wi-Fi module to transmit the monitoring data, and then display the data in the ThingSpeak IoT platform. However, to the author’s knowledge, no study reports on DO control to fulfill oxygen demand in the system [18], while another study developed real-time DO, pH, and temperature sensors using CC3200 Launchpad as the microcontroller and integrated with the Internet of Things (IoT) [19]. The real-time data from the study is displayed on the Node-RED dashboard.

Similarly, others reported the development of aquaculture monitoring systems consisting of pH, turbidity, and temperature sensors with the Things IoT platform and “if this then that (IFTTT)” integrated into the system to send a notification to the owner’s registered number [20]. A study was also conducted with the Thingspeak IoT platform for monitoring physicochemical variables like DO, pH, temperature, and salinity on fish farms in the Mekong Delta. This study also proposes automatic sensor probe cleaning to improve sensor reliability and reduce maintenance costs [21]. In another study, by building a system to monitor DO, pH, ammonia nitrogen, and temperature in real time, a warning notification was set to the user if the parameter was beyond the normal scope of the setting value [22]. However, the studies are only focused on monitoring and control for the aquaculture stock, and no study investigates the effect of monitoring and control on the biofilm so that good water quality can be achieved. Therefore, this study focuses on the development of a laboratory-scaled RAS monitoring and control system integrated with the Thingsboard IoT platform through Message Queuing Telemetry Transport (MQTT) architecture to allow investigation of the biofilm growth process and investigate the ammonia removal efficiency of the MBBR biofilter.

## 2. Design of Control and Monitoring System

This study focuses on the development of a laboratory-scale RAS equipped with IoT water quality monitoring and a controlled aerator to gain effectiveness and efficiency in TAN removal. Furthermore, the information about the diffuser mechanism and performance to increase oxygen levels, and its influence on the biofilm growth process, can be used to scale the system for future development. The laboratory-scale system in the experiment (Figure 1) consists of a biofilter (MBBR) tank, a fish tank, a bubble diffuser, an air pump, a water pump for water recirculation, and monitoring devices. The details of the monitoring system, as seen in Figure 2, consist of water quality sensors: DO, temperature, pH, TDS, and flow sensors. The system is also equipped with a microcontroller, Arduino Mega 2560, which functions as a data acquisition and processing device supplied by a power supply. In addition, the system has WiFi module ESP 8266-01 as the transmitter device and the IoT platform, Thingsboard, to display the monitoring data. The analog water quality sensors were connected to the corresponding signal converter and then arranged in parallel to communicate with the microcontroller for data acquisition, whereas, for the digital temperature sensor, a 4.7 kΩ resistor was added between the digital signal pin (blue line in Figure 2) and the 5 VDC power source to stabilize the temperature reading. For the digital water flow meter, the digital pin was directly connected to the microcontroller’s digital input. The output data from the sensors were read by the microcontroller and sent to the cloud using the WiFi module. Subsequently, the monitoring data was displayed on the Thingsboard website in real time. In addition, the system consists of a control element, relay, and air pump to ensure sufficient DO in the RAS.

### 2.1. Sensor Devices

#### 2.1.1. DO Sensor

DO is the free and non-compound oxygen level in water or other liquids [23]. In addition, DO is an important parameter that indicates water quality and reflects a biological, chemical, and physical process that occurs in the water [24]. Aquatic organisms in RAS, such as fish and bacteria, require DO to carry out biochemical processes such as respiration and metabolism. A sufficient amount of dissolved oxygen is necessary to ensure the efficient growth of aquatic life. When biofilm growth is efficient, the DO concentration in the system is above 4 mg/L and below 8 mg/L [25,26]. Thus, in this study, a galvanic-type sensor (DFRobot SKU:SEN0237) was used in the instrumentation system to observe the DO in real time. The galvanic type consists of an anode as the reference electrode and a cathode as a working electrode. Both the electrodes are inside a single electrode body containing electrolyte solution and separated from the measured medium by a non-conducting membrane partially permeable to oxygen [26]. When the DO sensor enters the water, the anode is oxidized and releases electrons, and the cathode undergoes reduction when an electron passes over. The oxidation and reduction are written in Equations (1)–(4) [18].
(1)$$\mathrm{Anode}=2Zn\to {2Zn}^{2+}+4{\;e}^{-}$$
(2)$$\mathrm{Cathode}\;\mathrm{reduction}\;\mathrm{reaction}=O_{2}+4{\;e}^{-}\;+{2H}_{2}O\to {4OH}^{-}$$
(3)$${4OH}^{-}+{2Zn}^{2+}\to 2Zn{\left(OH\right)}_{2}$$
(4)$$2Zn+O_{2}+{2H}_{2}O\to 2Zn{\left(OH\right)}_{2}$$

#### 2.1.2. pH Sensor

The pH (power of hydrogen) is an index of hydrogen ion concentration $\left(\left[H^{+}\right]\right)$ in the water on a scale of 0 to 14. The pH concept has its basis in ionization, as written in Equation (1) [27]. The optimum pH range for nitrification is between 7.0 and 9.0, whereas for nitrification bacteria growth like Nitrosomonas, the optimum pH range is between 7.2 and 8.8, and for Nitrobacter is between 7.2 and 9.0 [28]. A DFRobot Analog pH Pro Meter was used to ensure the system’s pH. The pH sensor ranges between 0 and 14 with a 0–60 °C temperature and accuracy ± 0.1 pH (@25 °C). The sensor consists of a sensing and reference electrode to read the potential difference between the sensing and reference electrodes [18].

#### 2.1.3. Temperature Sensor

Temperature is essential in wastewater treatment because temperature changes will influence the pH and DO [29,30]. Moreover, the temperature is a crucial factor in determining the growth of nitrifying bacteria. The optimal temperature range for nitrification in wastewater is 25–28 °C; nevertheless, *Nitrobacter* is more sensitive to environmental conditions than Nitrosomonas [31]. DS18B20 was used for the measurement range of −55 °C to 125 °C and accuracy of ±0.5 °C. The sensor consists of three wires: DQ for data communication, VDD for voltage, and GND for the ground.

#### 2.1.4. TDS Sensor

TDS are the amount of minerals, salts, organic matter, and metals dissolved in water and expressed in ppm (parts per million) or mg/L [32]. In RAS, dissolved solids consist of nitrogen and phosphorus compounds sourced from feeding or fertilizers. The increase in TDS will cause harmful nitrogen compounds that can cause stress to aquatic life. In the system, the DFRobot SKU:SEN0244 has a measurement range of 0–1000 ppm and an accuracy of ±10% FS (percentage of full scale) at 25 °C.

#### 2.1.5. Water Flow Sensor

A water flow sensor was used to ensure the proper flow rate for water recirculation so that there was no overflow in the biofilter. The flow rate converts to rpm to control the recirculation pump from the fish tank to the biofilter. In this study, a YF-201 was used as the flow sensor, which applies the Hall effect in the operation. When water passes through the sensor, it makes the turbine wheel rotate, creating a magnetic flux and interfering with the Hall effect sensor to produce a pulse signal output. This sensor has a 1–30 L/min flow range with a working voltage of 4.5–24 VDC.

### 2.2. Hardware Description of Control and IoT System

#### 2.2.1. Hardware Description of Control System

The control system utilized a relay that opens and closes a switch due to an input signal applied to the coil. The normally closed (NC) relay was connected to an AC source, and the relay’s COM pin was connected to the aerator pump. The input signal is the DO read from Arduino. When the DO is more than 8 mg/L, Arduino sends the signal to the relay, activates the magnet, and pulls the contact apart to open the circuit. When the DO is less than 6 mg/L, the electromagnet is energized and then attracts the contact towards the NC terminal so the current flows through the circuit. The control flow chart of the system can be seen in Figure 3.

#### 2.2.2. Hardware Description of IoT System

The ESP-01 Wi-Fi module allows Arduino to connect to the Wi-Fi network. Figure 4 describes the algorithm on Arduino for sending data to the Thingsboard. The library was included to provide functionality, and then the variable of the water quality sensor, DO control value, WiFi access, and the configuration access of the Thingsboard was defined and initialized. The sensor measurement data was read by Arduino and sent to the ThingsBoard IoT through Wi-Fi AP every 1 min automatically without external interruption.

The open-source Thingsboard IoT application program interface (API) provides an IoT platform to store and retrieve data from the DO, temperature, pH, TDS, and flow sensor using the MQTT publish–subscribe network protocol. As shown in Figure 5, the MQTT consists of three main participants in the message exchange process: publisher, MQTT broker, and subscriber [33]. The publisher is the device that produces the message, whereas, in this experiment, the sensors and Arduino acted as the publisher, whereas the ThingsBoard IoT platform serves as a broker and subscriber, which enables the message flow and then receives the message.

#### 2.2.3. Sensor Characteristics

The experiment to obtain sensor characteristics was conducted by comparing the sensors with handheld sensors with better accuracy. The characteristics of sensors include accuracy, precision, and error. The accuracy of the DO sensor was compared with a Milwaukee Mi 605 DO meter, which has ±1.5% FS accuracy and 0.01 mg/L resolution. The DO was measured at a flow rate of 0.4 L/min, according to the flow rate used. The measurement was conducted in a biofilter and fish tank to represent the experiment. The same measurement points were also used to test the rest of the water quality sensors. For the pH sensor, the experiment was performed by comparing the sensor with a Milwaukee Mi 101 pH meter with ±0.02 pH accuracy and 0.01 pH resolution, while the TDS sensor was compared with a Milwaukee EC59 ±2% FS accuracy and 1 µS/cm/1 ppm resolution, and the temperature sensor was compared with a Technol seven D617. All the sensor measurement data was collected every 10 s for 2 min. The accuracy, precision, error, standard deviation, and bias were calculated using Equations (A1)–(A5) (in Appendix A) [34].

## 3. MBBR Performance Evaluation

### 3.1. TAN Removal and Biofilm Thickness

The lab-scale RAS consists of a fish tank and MBBR with working volumes of 24 and 15 L. The water is circulated from the fish tank to MBBR by a water pump. K1 kaldness was used as the biofilm media, with a filling volume of 40% MBBR volume. A bubble diffuser provided the aeration with bubbles of an average diameter of 555 nm. At the beginning of the experiment, 1.2 g of Nitrobacter and Nitrosomonas bacteria was dissolved in the MBBR as the bacteria starter. The biofilm thickness was measured every 7 days using the gravimetric method [35].

The synthesis waste generated from 0.151 g of NH_4_Cl and 1.51 g of glucose produced 2 mg/L of TAN/day. The TAN concentration was measured using an API test kit every day. Meanwhile, the TAN removal efficiency was calculated from the reduced TAN concentration in 1 day.

### 3.2. Dynamic Measurement of $k_{L}a_{\left(T\right)}$

The performance of the bubble diffuser was determined by the ability of the diffuser to transfer the oxygen in the water, which can be distinguished from the $k_{L}a_{\left(T\right)}$ (volumetric mass transfer coefficient). The value can be derived from Equation (5), which is the mass balance for DO in the well-mixed liquid. From the equation, dC/dt is the accumulation of oxygen rate in the liquid phase, OTR is the oxygen transfer rate from gas to the liquid, and OUR is the oxygen uptake rate by the microorganisms.
(5)$$\frac{dC}{dt}=\mathrm{OTR}-\mathrm{OUR}$$

One method to measure DO transfer in water is using a dynamic method based on the absorption or desorption of oxygen. After the change in the inlet, the system results in a dynamic change in the DO concentration. The integration of Equation (1), where OUR = 0 was used to analyze the dynamic change in DO concentration, is written in Equation (6). From the equation, $k_{L}a_{\left(T\right)}$ is the oxygen transfer volumetric (min^−1^), C* is the DO saturation concentration, C_1_ is the DO at time t_1_, and C_2_ is the DO at time t_2_.
(6)$$\ln\;\left(\frac{C^{*}-C_{2}}{C^{*}-C_{1}}\right)=-k_{L}a_{\left(T\right)}\left(t_{2}-t_{1}\right)$$

The dynamic method involves desorption, where oxygen is supplied until it reaches saturation. Then, oxygen is eliminated from the liquid using nitrogen until the DO concentration is above the critical concentration of 0.5 mg/L. The other step in the dynamic method is absorption, where air is in supply until the DO saturation is reached [36,37]. The desorption and absorption methods can be expressed in Equations (7) and (8), respectively, where $C_{L}$ is the DO concentration at the time, t. In both steps, $k_{L}a_{\left(T\right)}$ can be determined from the slope of the ln f($C_{L}$) vs. the time graph.
(7)$$\ln\;\left(\frac{C^{*}}{C_{L}}\right)=k_{L}a_{\left(T\right)}t$$
(8)$$\ln\;\left(\frac{C_{L}}{C^{*}}\right)=-k_{L}a\;t$$

## 4. Result and Discussion

The monitoring of DO, temperature, pH, and TDS was conducted through the Thingsboard IoT platform, where the display can be accessed via the dashboard menu on the Thingsboard. The vertical y-axes of the graph on the Thingsboard show measurement data such as DO concentration in mg/L, pH, temperature in °C, and TDS in mg/L. At the same time, the horizontal x-axis shows the time interval. The data in Thingsboard are displayed in real-time but can also be accessed to correspond with the required period. Figure 6 shows the monitoring display data in 1 day with a grouping interval of 5 min. The monitoring data from Thingsboard can be converted into an Excel file and downloaded for analysis.

### 4.1. Water Quality Sensor Characteristic

The water quality sensor characteristics are listed in Table 1. The DO sensor measurement on the biofilter and fish tank resulted in good accuracy and precision, with an error of <5%. Likewise, the accuracy and precision of the pH sensor in the biofilter were high, with an error of ±0.84. The value represents the sensor consistency in the measurement. Meanwhile, an error of >5% was generated from the pH sensor in the fish tank, but the accuracy of the sensor was still within the permissible limits for water quality measurements (<10%) [38].

### 4.2. Result of the $k_{L}a_{\left(T\right)}$ Measurement

The measurement of $k_{L}a_{\left(T\right)}$ was carried out without the presence of bacteria in suspension before the bacterial growth experiment. The system can be used for $k_{L}a_{\left(T\right)}$ measurement because the measuring DO concentration data is sent every 1 min, so the $k_{L}a_{\left(T\right)}$ can be analyzed more accurately. The plot and result of the $k_{L}a_{\left(T\right)}$ measurement can be seen in Figure 7 and Table 2. The resulting $k_{L}a_{\left(T\right)}$ desorption from the experiment is 0.0068 min^−1^, while for $k_{L}a_{\left(T\right)},$ it is 0.0154 min^−1^. The high value of $k_{L}a_{\left(T\right)}$ indicates that a large oxygen transfer means an increase in DO concentration can be achieved faster [37,39].

In the experiment on biofilm growth, $k_{L}a_{\left(T\right)}$ was measured every week with the bacteria inside the suspension or liquid. As can be seen in Table 2, the $k_{L}a_{\left(T\right)}$ decreased compared with the $k_{L}a_{\left(T\right)}$ measured in liquid without bacteria. This happens because bacteria in liquid influence the liquid resistance, consequently decreasing the oxygen transfer. Aerobic bacteria act as interfacial blanketing that reduces the contact between gas–liquid interfaces [40]. The $k_{L}a_{\left(T\right)}$ increased in week 2 because the bacteria began to attach to the surface of the biofilm media. However, the $k_{L}a_{\left(T\right)}$ decreased in week 3 because the high TAN concentration in liquid increased resistance in the gas–liquid interface. The highest $k_{L}a_{\left(T\right)}$ was observed in week 5. This was related to the optimum biofilm performance, indicating that the TAN in the system can be degraded, causing the gas–liquid resistance to decrease.

### 4.3. Control System

Figure 8 shows the system with an aeration pump on–off control. The aeration pump turns off when it reaches the DO value of 8 mg/L, and then drops to 5.5 mg/L before turning on automatically. The rise time of the system was 7.36 mg/L when the aeration pump was turned on. The value represents how fast the system reaches the high value. The time needed by the system to reach a steady state was 2 min. The steady state of the system was maintained at around 7.3 mg/L.

### 4.4. Water Quality vs. Film Thickness

The experiment was conducted for 5 weeks to investigate the water quality monitoring and control of biofilm growth and TAN removal. The measurement data are plotted in Figure 9. In the first week, the MBBR showed good performance in TAN reduction. The highest TAN removal in the first week reached 50% on days 2 and 6. The TAN reduction is influenced by the thickness of the biofilm; in the first week, the biofilm thickness reached 98 µm. Then, the TAN concentration did not decrease in week 3 and week 4, as shown in Figure 9a,b. The stagnation could be caused by slight increases in the biofilm thickness of 109.38 µm, 112.50 µm, and 119.88 µm for weeks 2, 3, and 4, correspondingly. In contrast, the TAN concentration decreased at the beginning of week 5. The TAN removal efficiency reached 50% on day 28th, with the TAN concentration reduced by 2 mg/L. The high removal efficiency was due to the stable thickness of the biofilm at 104 µm.

Biofilm growth occurs in a cycle, from reversible attachment to irreversible attachment, maturation, and detachment [40,41]. The reversible attachment stage of the biofilm growth happens in the first week. Bacteria begin to attach on the surface of K1 Kaldness media. In this stage, bacteria need nutrition, resulting in a TAN concentration reduction [42]. In weeks 2 and 3, the biofilm was in the irreversible attachment stage and the beginning of the maturation stage. In this stage, the bacteria produced an extracellular polymers substance (EPS) to support the structure of the biofilm. In this stage, the bacteria in the biofilm entered a period of starvation to increase the adhesion and surface hydrophobicity of the biofilm. The starvation period of bacteria leads to the TAN, which is a nutrient for bacteria in biofilms, not to be consumed [43,44]. In week 4, the biofilm enters the maturation stage, which results in maximum thickness. Nevertheless, the TAN did not decrease in week 5, whereas the TAN removal efficiency was reduced due to the biofilm detachment stage at the biofilm thickness of 119.88 µm. Higher TAN removal efficiency was achieved when the DO concentration ranged from 5 to 7.36 mg/L. The highest concentration DO of 8.53 mg/L resulted on day 19 but did not significantly affect the TAN removal. This result showed that the effect of oxygen concentration on biofilm growth, nitrification, was effective at the DO concentration of 2–8 mg/L [26].

## 5. Conclusions

The IoT water monitoring and control system has been developed to maintain water quality and ensure proper biofilm growth. The water quality sensor showed an excellent precision of >95%, except for the DO sensor in the fish tank. The accuracy result is mostly below 95% but is still within the permissible limit. The water quality, comprising the DO, pH, temperature, and TDS values, was within the allowable limit. Moreover, using an on–off control method, the DO concentration was maintained in a range of 5–7.8. Therefore, by providing adequate dissolved oxygen in the MBBR biofilter tank, the biofilm achieved an optimum thickness of 119.88 µm and consequently received an optimum performance in TAN degradation of 50%. The limitation of the present study is that only two cycles of growth and detachment stages were observed. More observation with more growth cycles and detachment stages is needed to investigate whether it is possible to keep the stability of the reversible attachment, irreversible attachment, maturation, and detachment stages over time.

This study presents insights into the comprehensive applications of an IoT water monitoring and control system in MBBR that farmers should adopt in recirculating aquaculture systems (RAS). The system enables real-time measurements, alerts, and adjustments in critical water quality parameters such as temperature, pH, dissolved oxygen, ammonia, nitrite, and nitrate to demonstrate improved TAN removal effectiveness. IoT technology can also reduce costs associated with operations and logistics such as labor, maintenance, calibration, and transportation by automating the monitoring and control tasks and providing remote access to the data and the system. The MBBR biofilter with the IoT water quality monitoring and control system can increase RAS productivity and profitability by improving farmed species’ growth, survival, health, and welfare. At the same time, it enables a reduction in environmental impact and the risk of disease outbreaks and losses. However, IoT technology also presents some challenges and limitations: it requires competent users and appropriate training, advanced devices and sensors, and support from telecommunications networks to ensure the quality, validity, and usability of the data and the system.

## Figures and Tables

**Figure 1 sensors-24-00494-f001:**
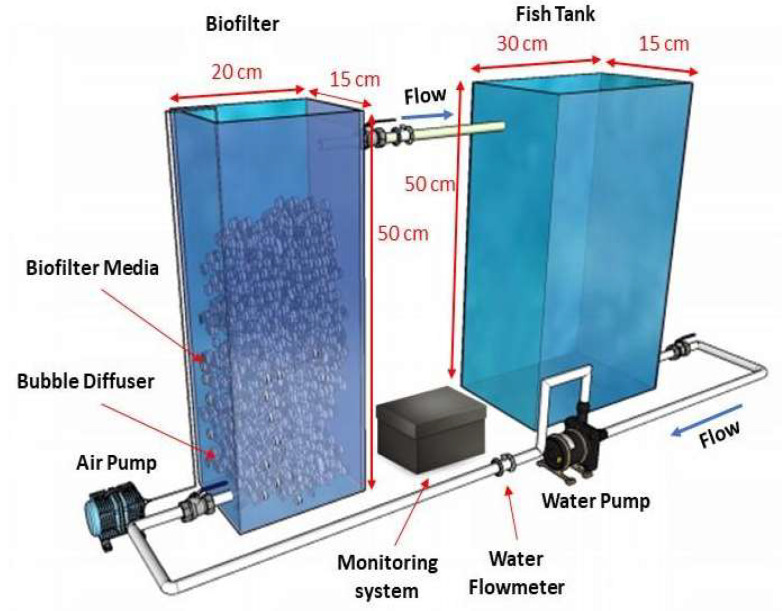
Laboratory-scale RAS.

**Figure 2 sensors-24-00494-f002:**
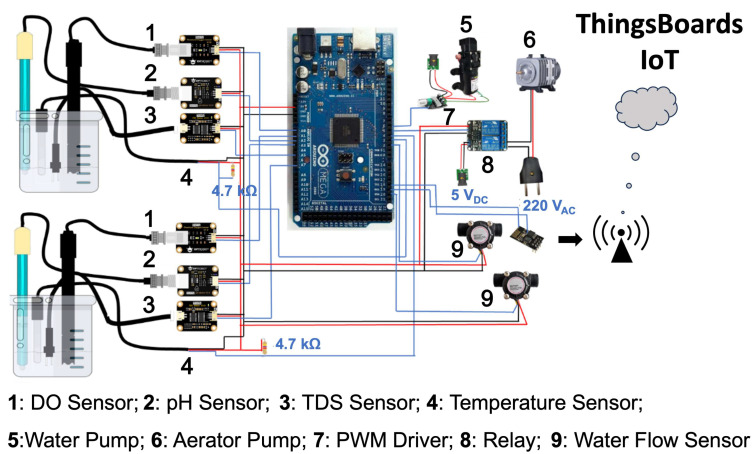
IoT monitoring and control system.

**Figure 3 sensors-24-00494-f003:**
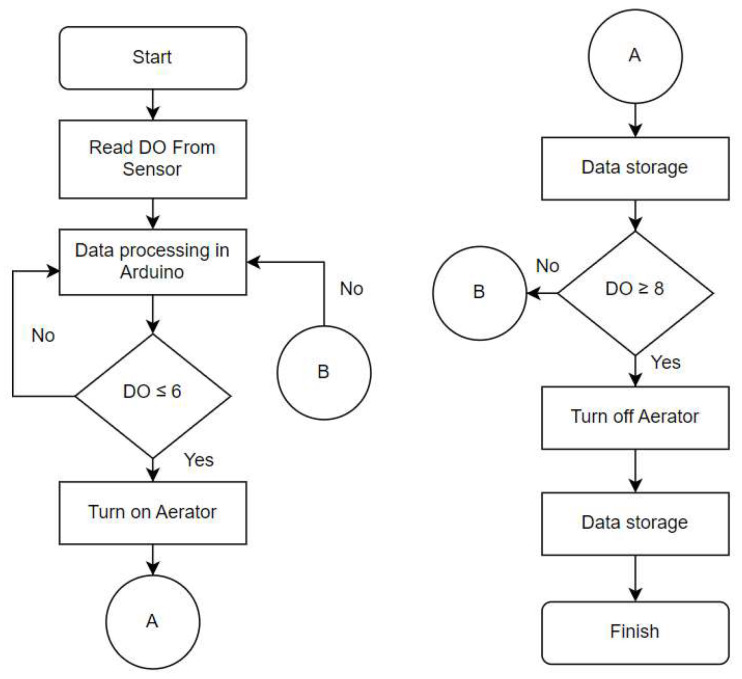
Aerator on–off control flowchart.

**Figure 4 sensors-24-00494-f004:**
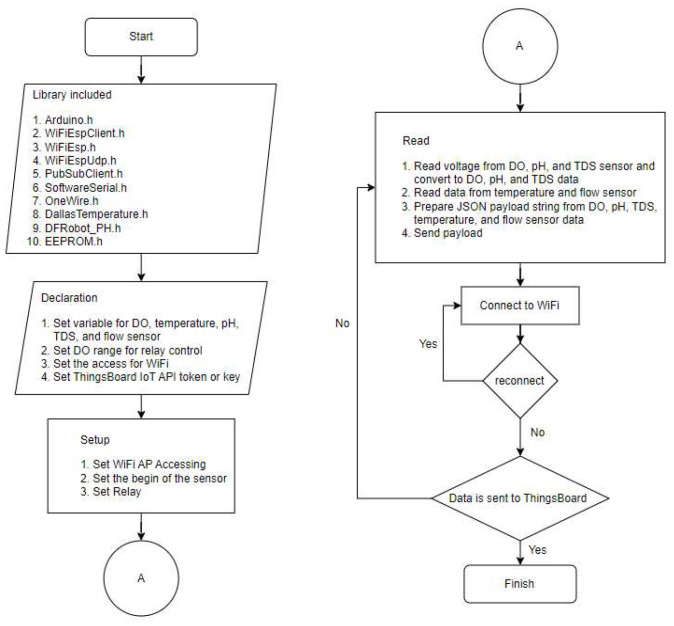
The algorithm of data acquisition and transmission to the Thingsboard IoT platform via Wi-Fi ESP-01.

**Figure 5 sensors-24-00494-f005:**
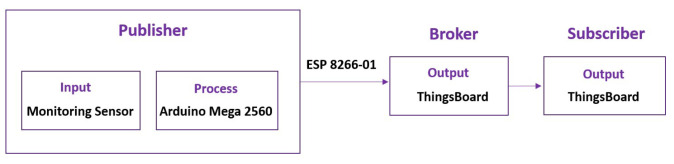
The components in the process of sending data using MQTT.

**Figure 6 sensors-24-00494-f006:**
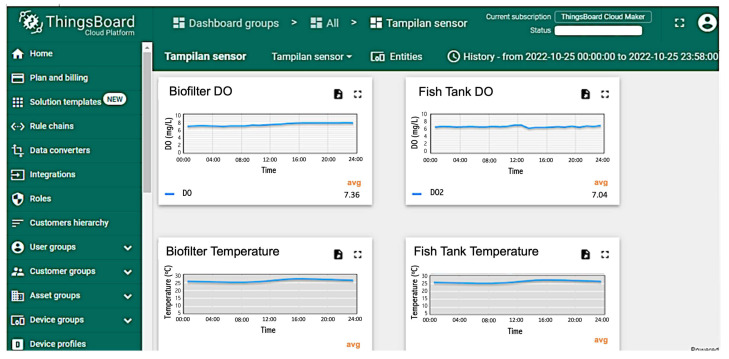
Thingsboard IoT platform display.

**Figure 7 sensors-24-00494-f007:**
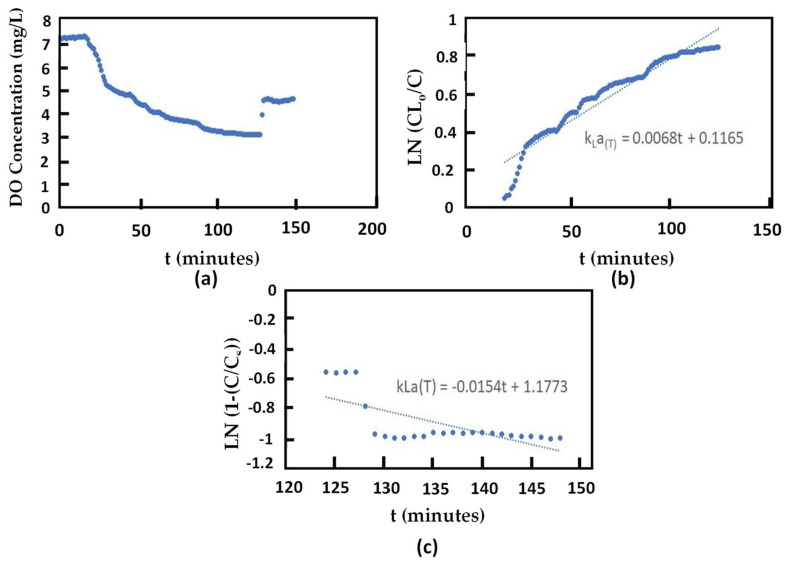
(**a**) The $k_{L}a_{\left(T\right)}$ measurement with gas out–gas in method; (**b**) $k_{L}a_{\left(T\right)}$ desorption plotting; (**c**) $k_{L}a_{\left(T\right)}$ absorption plotting.

**Figure 8 sensors-24-00494-f008:**
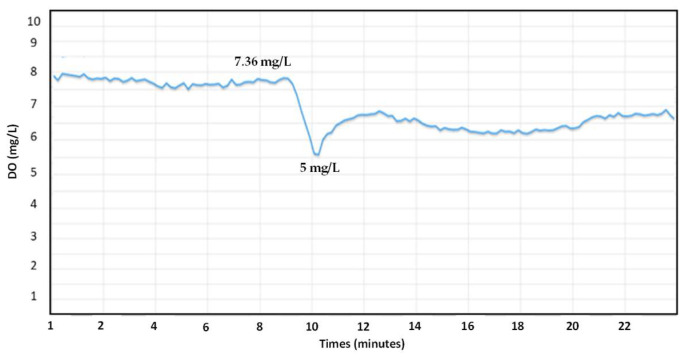
The DO concentration with on–off control.

**Figure 9 sensors-24-00494-f009:**
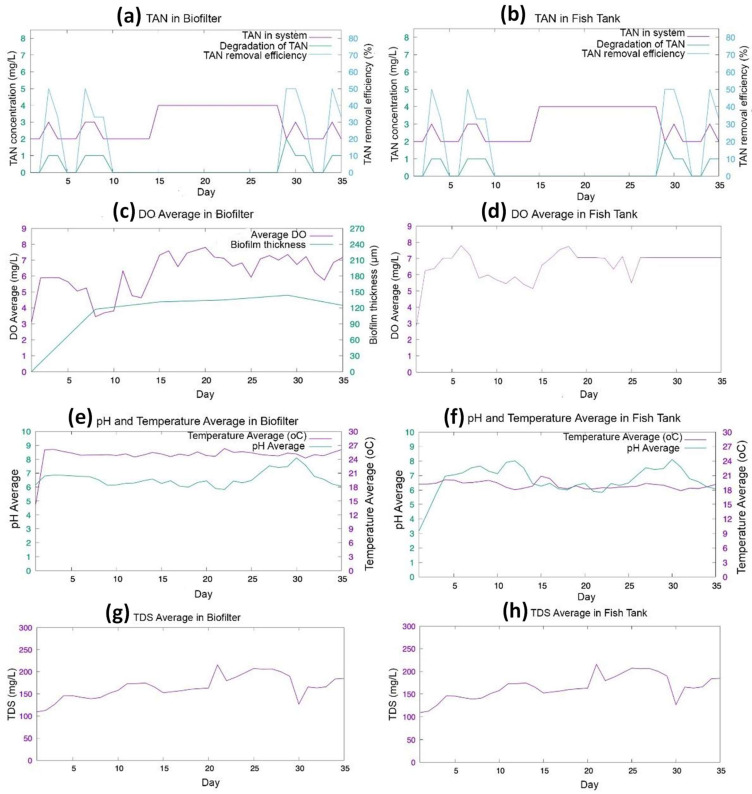
The TAN and water quality monitoring data: relation between total TAN, degradation of TAN, and TAN removal efficiency in (**a**) the biofilter; and (**b**) the fish tank; relation between DO concentration and biofilm thickness in (**c**) the biofilter; and (**d**) the fish tank; relation between pH and temperature in (**e**) the biofilter; and (**f**) the fish tank. TDS concentration in (**g**) the biofilter; and (**h**) the fish tank.

**Table 1 sensors-24-00494-t001:** The result of sensor characterization measurement.

Sensor	Place	Accuracy (%)	Precision (%)	Error (%)
DO	Biofilter	98.97	98.31	±1.03
	Fish tank	95.53	94.50	±4.47
pH	Biofilter	99.16	99.84	±0.84
	Fish tank	93.61	98.79	±6.39
Temperature	Biofilter	97.92	99.75	±2.08
	Fish tank	96.56	95.16	±3.44
TDS	Biofilter	92.83	98.81	±7.17
	Fish tank	93.26	98.74	±6.74

**Table 2 sensors-24-00494-t002:** The result of $k_{L}a_{\left(T\right)}$ measurement.

Week	$k_{L}a_{\left(T\right)}$ Coarse Bubble (min^−1^)	
	Desorption	Absorption
0 (without bacteria)	0.0042	0.0005
1	0.0099	0.0293
2	0.0261	0.0472
3	0.0099	0.0293
4	0.0143	0.0011
5	0.0144	0.0082

## Data Availability

Data are contained within the article.

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
