# Peer review of "IoT Water Quality Monitoring and Control System in Moving Bed Biofilm Reactor to Reduce Total Ammonia Nitrogen"

_sensors, 2024, doi:10.3390/s24020494_

Round 1

Reviewer 1 Report

Comments and Suggestions for Authors

The presented manuscript explores the latest and state-of-the-art progress in IoT Water Quality Monitoring and Control System in Moving Bed Biofilm Reactor to Reduce Total Ammonia Nitrogen. However, some issues require to be addresses scientifically. I recommend minor revisions.

v Please make sure to define each acronym at its first use. Check through the entire manuscript to make sure it is defined at the first use.

v Grammar needs to be improved significantly.

v The literature review needs to more update. It is need to update with recent related research: (Biosensors For Monitoring Heavy Metals Contamination In The Wastewater. Recent Advances in Biosensor Technology 2023, 203–211. https://doi.org/10.2174/9789815123739123010013)

v Need to update conclusion section

Comments on the Quality of English Language

v Grammar needs to be improved significantly.

Author Response

Response to Reviewer 1 Comments

Point 1: The presented manuscript explores the latest and state-of-the-art progress in IoT Water Quality Monitoring and Control System in Moving Bed Biofilm Reactor to Reduce Total Ammonia Nitrogen. However, some issues require to be addresses scientifically. I recommend minor revisions..

Response 1: We would like to thank to reviewer #1 to provide valuable comments and suggestions to improve the quality of the manuscript.

Point 2: Please make sure to define each acronym at its first use. Check through the entire manuscript to make sure it is defined at the first use.

Response 2: We have revised the manuscript to define each acronym at its first use and checked through the entire manuscript to ensure it is determined at the first use.

Point 3: Grammar needs to be improved significantly..

Response 3: We have improved the grammar in the entire manuscript.

Point 4: The literature review needs to more update. It is need to update with recent related research: (Biosensors For Monitoring Heavy Metals Contamination In The Wastewater. Recent Advances in Biosensor Technology 2023, 203–211. https://doi.org/10.2174/9789815123739123010013).

Response 4: We have updated the literature review by including the recommended reference in the revised version and highlighted in blue as follows:

The Internet of Things (IoT) has recently gained much interest in real-time data measurement for cost-effective aquaculture management. IoT refers to smart devices or sensors that are uniquely addressable based on their communication protocols, are adaptable, and are autonomous with inherent security [9]. IoT water quality monitoring uses intelligent sensors and wireless communication to measure and analyze various water quality parameters, such as temperature, pH, turbidity, dissolved oxygen, etc. IoT water quality monitoring has many benefits, such as real-time data collection, remote access, automation, cost-effectiveness, and scalability [10-12]. However, IoT water quality monitoring also faces many challenges and limitations, such as data security, communication coverage, energy consumption, sensor calibration, data quality, and interoperability [13,14]. Moreover, IoT water quality monitoring applications may have different requirements and constraints, such as water source, environment, types of sensors, purpose, and stakeholders [15,16]. Therefore, there is no one-size-fits-all solution for IoT water quality monitoring, and different methods, techniques, models, systems, and applications need to be compared and contrasted based on their strengths and weaknesses, as well as their suitability and feasibility for different scenarios.

Additional references:

  1. Zulkifli, C.Z.; Garfan, S.; Talal, M.; Alamoodi, A.H.; Alamleh, A.; Ahmaro, I.Y.Y.; Sulaiman, S.; Ibrahim, A.B.; Zaidan, B.B.; Ismail, A.R.; et al. IoT-Based Water Monitoring Systems: A Systematic Review. Water 2022, 14, 3621. https://doi.org/10.3390/w1422362
  2. Hemdan, E.ED.; Essa, Y.M.; Shouman, M.; El-Sayed, A.; Moustafa, A.N. An efficient IoT based smart water quality monitoring system. Multimed Tools Appl. 2023, 82, 28827–28851. https://doi.org/10.1007/s11042-023-14504-z
  3. Mustafa, H.M.; Mustapha, A.;  Hayder, G.; Salisu, A. Applications of IoT and Artificial Intelligence in Water Quality Monitoring and Prediction: A Review. In the 6th International Conference on Inventive Computation Technologies (ICICT), Coimbatore, India, 2021, pp. 968-975. https://doi.org/10.1109/ICICT50816.2021.9358675
  4. Hashemi, H.; Mirnasab, M.A. "Internet of Things": An Emerging Real-Time Technology for Environmental Health Monitoring. JEHSD 2018, 3(1), 432-5.
  5. Kumar, M.; Singh, T.; Maurya, M.K.; Shivhare, A.; Raut, A.; Singh, P.K. Quality Assessment and Monitoring of River Water Using IoT Infrastructure, IEEE Internet of Things Journal 2023, 10(12), 10280-10290. https://doi.org/10.1109/JIOT.2023.3238123
  6. Jan, F.; Min-Allah, N.; DüÅŸtegör, D. IoT Based Smart Water Quality Monitoring: Recent Techniques, Trends and Challenges for Domestic Applications. Water 2021, 13, 1729. https://doi.org/10.3390/w13131729
  7. Amra Odobašić, Indira Šestan and Sabina Begić, Biosensors For Monitoring Heavy Metals Contamination In The Wastewater Recent Advances in Biosensor Technology 2023, pp. 203–211. https://doi.org/10.2174/9789815123739123010013

Point 5:                 Need to update conclusion section.

Response 3: The conclusion has been updated and highlighted in blue as follows:

The IoT water monitoring and control system has been developed to maintain water quality and ensure proper biofilm growth. The water quality sensor showed an excellent precision >95%, except for the DO sensor in the fish tank. The accuracy result is mostly below 95% but is still within the permissible limit. The water quality, such as DO, pH, temperature, and TDS value, were within the allowable limit. Moreover, using an on-off control method, the DO concentration was maintained at a range of 5-7.8. Therefore, providing adequate dissolved oxygen in the MBBR biofilter tank, the biofilm achieved an optimum thickness of 119.88 µm and consequently received an optimum performance of TAN degradation of 50%. The limitation of the presence study is that only two cycles of growth and detachment stages were observed. More observation with few growth cycles and detachment stages is needed to investigate whether it is possible to keep the stability of reversible attachment, irreversible attachment, maturation, and detachment stages over time.

Here are insights into the comprehensive applications of the IoT water monitoring and control system in MBBR that farmers should adopt in Recirculating Aquaculture Systems (RAS). The system enables real-time measurements, alerts, and adjustments of critical water quality parameters such as temperature, pH, dissolved oxygen, ammonia, nitrite, and nitrate to demonstrate improved TAN removal effectiveness. IoT technology can also reduce costs associated with operations and logistics such as labor, maintenance, calibration, and transportation by automating the monitoring and control tasks and providing remote access to the data and the system. The MBBR biofilter with IoT water quality monitoring and control system can increase RAS productivity and profitability by improving farmed species' growth, survival, health, and welfare. At the same time, it enables the reduction of environmental impact and the risk of disease outbreaks and losses. However, IoT technology also presents some challenges and limitations: it requires competent users and appropriate training, advanced devices and sensors, and support from telecommunications networks to ensure the quality, validity, and usability of the data and the system.

Reviewer 2 Report

Comments and Suggestions for Authors

IoT Water Quality Monitoring and Control System in Moving Bed Biofilm Reactor to Reduce Total Ammonia Nitrogen Authors. Putu A. Suriasni  , Ferry Faizal  , Wawan Hermawan  , Ujang Subhan  , Camellia Panatarani  , I Made Joni * The proposed manuscript introduce set up and data from IoT Water Quality Monitoring and Control System in Moving Bed Biofilm Reactor to Reduce Total Ammonia Nitrogen. The referee suggests a check of the following aspects. 1. Introduction should have some piece of information about the state of art for systems or mechanisms that perform for water control quality with the same criterion (IoT water quality monitoring). The referee means some examples from literature. IoT is a common approach for environmental monitoring. 2. Check if some part of introduction can be moved away. According the referee part of the information reported, especially in the last part of introduction, can be moved in the next section. 3. Improve the quality of all Figures. 4. For accuracy authors compared values with Milwaukee M60SDO meter. Did the authors previously perform some chemical tests or calibration curves to corroborate/check data? 5. Check numbers of sections. There are some discrepancies in the numerical order (2.14 twice in the text). 6. If possible, the referee suggests to move some information in a supplementary section, with the aim to improve the presentation of the scientific work.  7. All acronyms must be definde the first time they are reported in a manuscript. Please check this aspect.      

Comments on the Quality of English Language

Check if it is possible to improve the quality of English

Author Response

Response to Reviewer 2 Comments

Point 1: IoT Water Quality Monitoring and Control System in Moving Bed Biofilm Reactor to Reduce Total Ammonia Nitrogen Authors. Putu A. Suriasni, Ferry Faizal, Wawan Hermawan, Ujang Subhan, Camellia Panatarani, I Made Joni * The proposed manuscript introduce set up and data from IoT Water Quality Monitoring and Control System in Moving Bed Biofilm Reactor to Reduce Total Ammonia Nitrogen. The referee suggests a check of the following aspects.

Response 1: We would like to thank the referee for providing valuable comments and suggestions to improve the manuscript's quality.

Point 2: Introduction should have some piece of information about the state of art for systems or mechanisms that perform for water control quality with the same criterion (IoT water quality monitoring). The referee means some examples from literature. IoT is a common approach for environmental monitoring..

Response 2: We have revised the introduction section to include information about the state-of-the-art systems or mechanisms that perform for water control quality with the same criterion (IoT water quality monitoring). The revised version has been highlighted in blues as follows:

The Internet of Things (IoT) has recently gained much interest in real-time data measurement for cost-effective aquaculture management. IoT refers to smart devices or sensors that are uniquely addressable based on their communication protocols, are adaptable, and are autonomous with inherent security [9]. IoT water quality monitoring uses intelligent sensors and wireless communication to measure and analyze various water quality parameters, such as temperature, pH, turbidity, dissolved oxygen, etc. IoT water quality monitoring has many benefits, such as real-time data collection, remote access, automation, cost-effectiveness, and scalability [10-12]. However, IoT water quality monitoring also faces many challenges and limitations, such as data security, communication coverage, energy consumption, sensor calibration, data quality, and interoperability [13,14]. Moreover, IoT water quality monitoring applications may have different requirements and constraints, such as water source, environment, types of sensors, purpose, and stakeholders [15,16]. Therefore, there is no one-size-fits-all solution for IoT water quality monitoring, and different methods, techniques, models, systems, and applications need to be compared and contrasted based on their strengths and weaknesses, as well as their suitability and feasibility for different scenarios.

Additional references:

  1. Zulkifli, C.Z.; Garfan, S.; Talal, M.; Alamoodi, A.H.; Alamleh, A.; Ahmaro, I.Y.Y.; Sulaiman, S.; Ibrahim, A.B.; Zaidan, B.B.; Ismail, A.R.; et al. IoT-Based Water Monitoring Systems: A Systematic Review. Water 2022, 14, 3621. https://doi.org/10.3390/w1422362
  2. Hemdan, E.ED.; Essa, Y.M.; Shouman, M.; El-Sayed, A.; Moustafa, A.N. An efficient IoT based smart water quality monitoring system. Multimed Tools Appl. 2023, 82, 28827–28851. https://doi.org/10.1007/s11042-023-14504-z
  3. Mustafa, H.M.; Mustapha, A.;  Hayder, G.; Salisu, A. Applications of IoT and Artificial Intelligence in Water Quality Monitoring and Prediction: A Review. In the 6th International Conference on Inventive Computation Technologies (ICICT), Coimbatore, India, 2021, pp. 968-975. https://doi.org/10.1109/ICICT50816.2021.9358675
  4. Hashemi, H.; Mirnasab, M.A. "Internet of Things": An Emerging Real-Time Technology for Environmental Health Monitoring. JEHSD 2018, 3(1), 432-5.
  5. Kumar, M.; Singh, T.; Maurya, M.K.; Shivhare, A.; Raut, A.; Singh, P.K. Quality Assessment and Monitoring of River Water Using IoT Infrastructure, IEEE Internet of Things Journal 2023, 10(12), 10280-10290. https://doi.org/10.1109/JIOT.2023.3238123
  6. Jan, F.; Min-Allah, N.; DüÅŸtegör, D. IoT Based Smart Water Quality Monitoring: Recent Techniques, Trends and Challenges for Domestic Applications. Water 2021, 13, 1729. https://doi.org/10.3390/w13131729
  7. Amra Odobašić, Indira Šestan and Sabina Begić, Biosensors For Monitoring Heavy Metals Contamination In The Wastewater Recent Advances in Biosensor Technology 2023, pp. 203–211. https://doi.org/10.2174/9789815123739123010013

Point 3: Check if some part of introduction can be moved away. According the referee part of the information reported, especially in the last part of introduction, can be moved in the next section.

Response 3: The last part of the introduction has been moved to the next section and highlighted in blue as follows:

The development of laboratory-scale RAS equipped with IoT water quality monitoring and controlled aerator to gain effectiveness and efficiency of TAN removal. Furthermore, the information about the diffuser mechanism and performance to increase oxygen levels and its influence on the biofilm growth process can be used to scale the system for future development.

Point 4: Improve the quality of all Figures.

Response 4: We have improved the quality of all Figures.

Point 5:                 For accuracy authors compared values with Milwaukee M60SDO meter. Did the authors previously perform some chemical tests or calibration curves to corroborate/check data?.

Response 5: We always do a calibration of the Milwaukee M60SDO following the procedure recommended by the company using a simple 2 point open air calibration with MA9071 electrolyte solution to ensure that MA840 reinforced polarographic probe with an oxygen-permeable polytetrafluroethylene (PTFE) membrane sensor still worked properly.

Point 6:                 Check numbers of sections. There are some discrepancies in the numerical order (2.14 twice in the text).

Response 6: We have revised the numerical order of the section.

Point 7:                 If possible, the referee suggests to move some information in a supplementary section, with the aim to improve the presentation of the scientific work. 

Response 7: We have moved into a supplementary section or Appendix A for the calculation procedure on the accuracy, precision, error, standard deviation, and bias.

Point 8: All acronyms must be definde the first time they are reported in a manuscript. Please check this aspect.   

Response 8: All acronyms have been defined the first time they are reported in a manuscript.

Reviewer 3 Report

Comments and Suggestions for Authors

Dear Authors,

Thanks for your valuable research

1.      Is it cost benefit in full scale?

2.      Please write the limitation of this method in practice.

3.      Please use following related article in your paper: 

·        Hassan Hashemi, Mohammad Amin Mirnasab. "Internet of Things": An Emerging Real-Time Technology for Environmental Health Monitoring. JEHSD, Vol (3), Issue (1), March 2018, 432-5

Best Regards,

Author Response

Response to Reviewer 3 Comments

Point 1: Thanks for your valuable research. Is it cost benefit in full scale?

Response 1: We have added insight into the cost benefits in the full-scale applications at the end of the conclusion:

Here are insights into the comprehensive applications of the IoT water monitoring and control system in MBBR that farmers should adopt in Recirculating Aquaculture Systems (RAS). The system enables real-time measurements, alerts, and adjustments of critical water quality parameters such as temperature, pH, dissolved oxygen, ammonia, nitrite, and nitrate to demonstrate improved TAN removal effectiveness. IoT technology can also reduce costs associated with operations and logistics such as labor, maintenance, calibration, and transportation by automating the monitoring and control tasks and providing remote access to the data and the system. The MBBR biofilter with IoT water quality monitoring and control system can increase RAS productivity and profitability by improving farmed species' growth, survival, health, and welfare. At the same time, it enables the reduction of environmental impact and the risk of disease outbreaks and losses. However, IoT technology also presents some challenges and limitations: it requires competent users and appropriate training, advanced devices and sensors, and support from telecommunications networks to ensure the quality, validity, and usability of the data and the system.

Point 2: Please write the limitation of this method in practice.

Response 2: The limitations of the method in practice have been added in the revised version of the manuscript in the conclusion as follows:

The limitation of the presence study is that only two cycles of growth and detachment stages were observed. More observation with few growth cycles and detachment stages is needed to investigate whether it is possible to keep the stability of reversible attachment, irreversible attachment, maturation, and detachment stages over time.

Point 3: Please use following related article in your paper: Hassan Hashemi, Mohammad Amin Mirnasab. "Internet of Things": An Emerging Real-Time Technology for Environmental Health Monitoring. JEHSD, Vol (3), Issue (1), March 2018, 432-5.

Response 3: We have used the suggested reference in the revised version of the manuscript.
